# Analytical framework to evaluate and optimize the use of imperfect diagnostics to inform outbreak response: Application to the 2017 plague epidemic in Madagascar

Quirine ten Bosch[1,2☯]*, Voahangy Andrianaivoarimanana[3☯], Beza Ramasindrazana[3☯], Guillain Mikaty[4], Rado J. L. Rakotonanahary[3], Birgit Nikolay[1], Soloandry Rahajandraibe[3], Maxence Feher[4], Quentin Grassin[4], Juliette Paireau[1], Soanandrasana Rahelinirina[3], Rindra Randremanana[5], Feno Rakotoarimanana[5], Marie Melocco[5], Voahangy Rasolofo[6], Javier Pizarro-Cerdá[7,8,9], Anne-Sophie Le Guern[7,8,9], Eric Bertherat[10], Maherisoa Ratsitorahina[6,11], André Spiegel[6], Laurence Baril[5☯], Minoarisoa Rajerison[3☯], Simon Cauchemez[1☯]

1 Mathematical Modelling of Infectious Diseases Unit, Institut Pasteur, Université Paris Cité, CNRS UMR2000, F-75015 Paris, France, 2 Quantitative Veterinary Epidemiology, Department of Animal Sciences, Wageningen University and Research, Wageningen, the Netherlands, 3 Plague Unit, Institut Pasteur de Madagascar, Antananarivo, Madagascar, 4 Environment and Infectious Risks Research Unit, Laboratory for Urgent Response to Biological Threats (ERI-CIBU), Institut Pasteur, Paris, France, 5 Epidemiology and Clinical Research Unit, Institut Pasteur de Madagascar, Antananarivo Madagascar, 6 Direction, Institut Pasteur de Madagascar, Antananarivo, Madagascar, 7 Yersinia Research Unit, Institut Pasteur, Université Paris Cité, CNRS UMR 6047, F-75015 Paris, France, 8 National Reference Laboratory for Plague and other Yersiniosis, Institut Pasteur, F-75015 Paris, France, 9 World Health Organization Collaborating Center for Plague FRA-140, Institut Pasteur, F-75015 Paris, France, 10 World Health Organization, Health Emergency Programme, Department of Infectious Hazard Management, Geneva, Switzerland, 11 Directorate of Health and Epidemiological Surveillance, Ministry of Public Health, Antananarivo, Madagascar

☯ These authors contributed equally to this work.
* quirine.tenbosch@wur.nl

**Data Availability Statement:** Data and code required to reproduce the analysis in this

## Abstract

During outbreaks, the lack of diagnostic "gold standard" can mask the true burden of infection in the population and hamper the allocation of resources required for control. Here, we present an analytical framework to evaluate and optimize the use of diagnostics when multiple yet imperfect diagnostic tests are available. We apply it to laboratory results of 2,136 samples, analyzed with 3 diagnostic tests (based on up to 7 diagnostic outcomes), collected during the 2017 pneumonic (PP) and bubonic plague (BP) outbreak in Madagascar, which was unprecedented both in the number of notified cases, clinical presentation, and spatial distribution. The extent of these outbreaks has however remained unclear due to nonoptimal assays. Using latent class methods, we estimate that 7% to 15% of notified cases were *Yersinia pestis*-infected. Overreporting was highest during the peak of the outbreak and lowest in the rural settings endemic to *Y. pestis*. Molecular biology methods offered the best compromise between sensitivity and specificity. The specificity of the rapid diagnostic test was relatively low (PP: 82%, BP: 85%), particularly for use in contexts with large quantities of misclassified cases. Comparison with data from a subsequent seasonal *Y. pestis* outbreak in 2018 reveal better test performance (BP: specificity 99%, sensitivity: 91%), indicating that

manuscript are available on Open Science Framework: https://osf.io/nbc4t/.

**Funding:** This work was supported by Wellcome Trust/ Department of International Development (Grant 211309/Z/18/Z; https://wellcome.org/ supporting MM, RR, and FMR), AXA Research Fund (https://www.axa-research.org/ supporting QTB, BN, JP and SC), and the Laboratoire d'Excellence Integrative Biology of Emerging Infectious Diseases program (Grant ANR-10-LABX-62-IBEID; https://anr.fr/ supporting JPC, QTB, BN, JP and SC). The funders had no role in study design, data collection and analysis, decision to publish, or preparation of the manuscript.

**Competing interests:** The authors have declared that no competing interests exist.

**Abbreviations:** BP, bubonic plague; CFR, case fatality ratio; CLP, central laboratory for plague; cPCR, classical polymerase chain reaction; IPM, Institut Pasteur de Madagascar; MB, molecular biology; MCMC, Markov chain Monte Carlo; PP, pneumonic plague; PPV, positive predictive value; qPCR, quantitative polymerase chain reaction; RDT, rapid diagnostic test.

factors related to the response to a large, explosive outbreak may well have affected test performance. We used our framework to optimize the case classification and derive consolidated epidemic trends. Our approach may help reduce uncertainties in other outbreaks where diagnostics are imperfect.

## Introduction

The availability of accurate diagnostics is essential for an effective response to infectious disease outbreaks. In the relatively common situation where no gold standard diagnostic is available (i.e., absence of a diagnostic test with perfect sensitivity and specificity), interpretation of diagnostic results becomes challenging [1,2]. This may hamper case identification and management; jeopardize the evaluation of the burden, scope, timing, and spatial expansion of the outbreak; and ultimately impede control. Here, taking a large plague outbreak in Madagascar as a case study, we present an integrative analytical framework to assess the performance of diagnostics and reconstruct spatiotemporal epidemic patterns in situations where multiple yet imperfect diagnostics are available.

Plague is a highly fatal disease caused by a gram-negative bacillus *Yersinia pestis* [3]. Rodents constitute its natural reservoir and the bacillus can be transmitted to humans by fleas. When bitten by an infected flea, a person typically develops bubonic plague (BP), which is characterized by fever and painful lymphadenitis in the area of the fleabite [3]. Septicemic spread can occasionally lead to pneumonic plague (PP) that typically consists of sudden fever, cough, and symptoms of lower respiratory tract infections. Interhuman transmission of PP is possible through droplet spread [4]. Plague case fatality ratio (CFR) has been estimated between 10% to 40% [5–7]. Diagnosis, particularly of PP, is challenging due to (i) nonspecific early symptoms [8,9]; (ii) the difficulty to collect high-quality sputum samples, especially from severely ill and young patients [10]; and (iii) the scarcity of PP cases hampering evaluation of diagnostics; most assays have been evaluated on BP samples [11].

Between August and November 2017, Madagascar experienced a large number (2,414) of notifications of clinically suspected plague cases that were predominantly in 2 major urban areas (79%) with unusually high proportions of PP (78%) (Fig 1A and 1B) [12]. Important discrepancies between tests (the proportion of positive PP results ranged from 1% to 18% depending on the test; Fig 1C and 1D) mean that the true extent of the PP outbreak remains unclear. Besides, without a good understanding of the performances of the diagnostics available, it is difficult to optimize diagnostic and case classification algorithms for future outbreaks. Here, we analyze data describing this large plague epidemic to obtain a comprehensive view of the burden of infection among notified cases. We evaluate the performance of test diagnostics and propose updated case classification algorithms to better allocate sparse resources during future outbreaks. Using the combined test results and diagnostic performance estimates, we reconstruct epidemiological trends over space and time.

## Results

Of 2,414 notifications, we consider those with sputum or bubo aspirates and known clinical form (PP: 1,779, BP: 357) [12]. Of PP sputum samples, 22% have at least 1 positive culture ($N = 4$), rapid diagnostic test (RDT) ($N = 327$), or molecular biology (MB) ($N = 84$) (Fig 1D) and are classified, based on their diagnostic outcomes (Fig 2), as either confirmed (2%) or probable (20%) (Fig 1C), versus 34% of BP (37 culture, 99 RDT, 79 MB) (Fig 1D) with 16% confirmed and 18% probable (Fig 1C) [12].

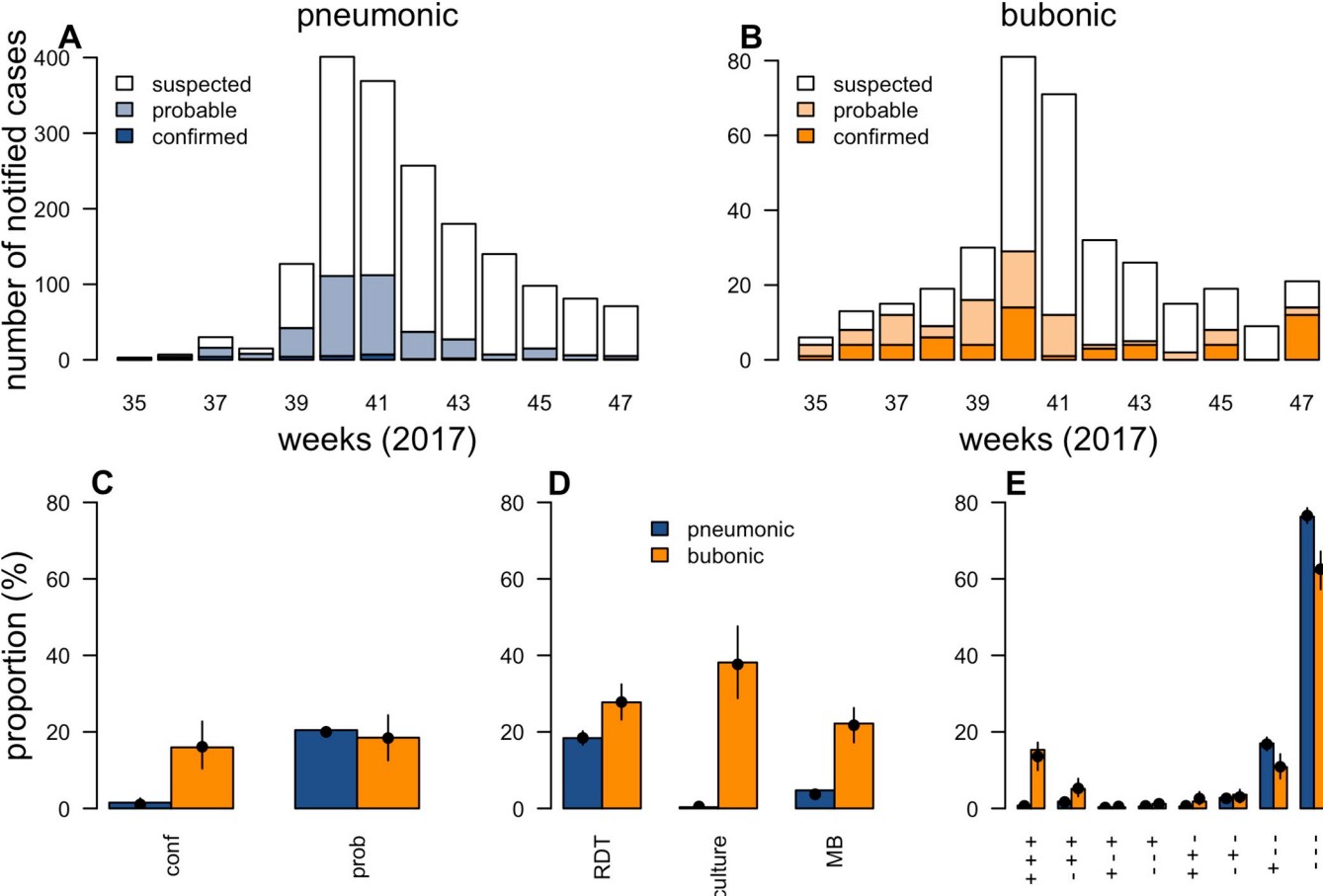

**Fig 1. Diagnostics and case classification during the plague outbreak in Madagascar in 2017.** (A, B) Weekly number of notified cases for PP (A) and BP (B) by case classification. (C–E) Proportion of notified cases classified as confirmed (conf) or probable (prob) (C), with a positive test result for RDT, culture, or MB (NB, only cases on whom the respective test was performed are considered in the denominator. No restrictions were put on the use of MB and RDT. Culture was only performed if RDT was positive, apart from PP samples from nonendemic regions. On those samples, culture was performed irrespective of RDT result) (D) and with a certain combination of diagnostic outcomes (E), presenting outcomes that were performed on all samples (RDT, qPCR on *pla* and *caf1* genes). Model fits to these proportions are provided with black dots and lines indicating model predictions and 95% credible intervals, respectively. The underlying data and code to reproduce this figure are available on Open Science Framework (https://osf.io/nbc4t/). BP, bubonic plague; MB, molecular biology; PP, pneumonic plague; qPCR, quantitative polymerase chain reaction; RDT, rapid diagnostic test.

We develop a latent-class statistical model [13] to estimate the performance of diagnostic tests and the scale of the outbreak from contingency tables describing 3 tests with up to 7 separate diagnostics outcomes (i.e., 2 single-outcome tests: RDT, culture; plus up to 5 genes for MB) for 2,136 samples received at the central laboratory for plague (CLP) between August 1 and November 26, 2017. The model describes the joint expected distribution of diagnostic outcomes as a function of the prevalence (proportion of *Y. pestis* infections among notified, clinically suspected cases), the sensitivity (probability of positive result if the sample is from a *Y. pestis*-infected person), and specificity (probability of negative result if the sample is from a person that was not infected with *Y. pestis*) of each test. Estimation of model parameters is performed in a Bayesian framework via Markov chain Monte Carlo (MCMC) sampling [14] under the assumption that culture specificity is 100%. Technical details are provided in Materials and methods.

We estimate that test specificity was similar between sample types. MB was highly specific (PP: 100%, 95% credible interval 99% to 100%, BP: 100%, 98% to 100%), whereas RDT

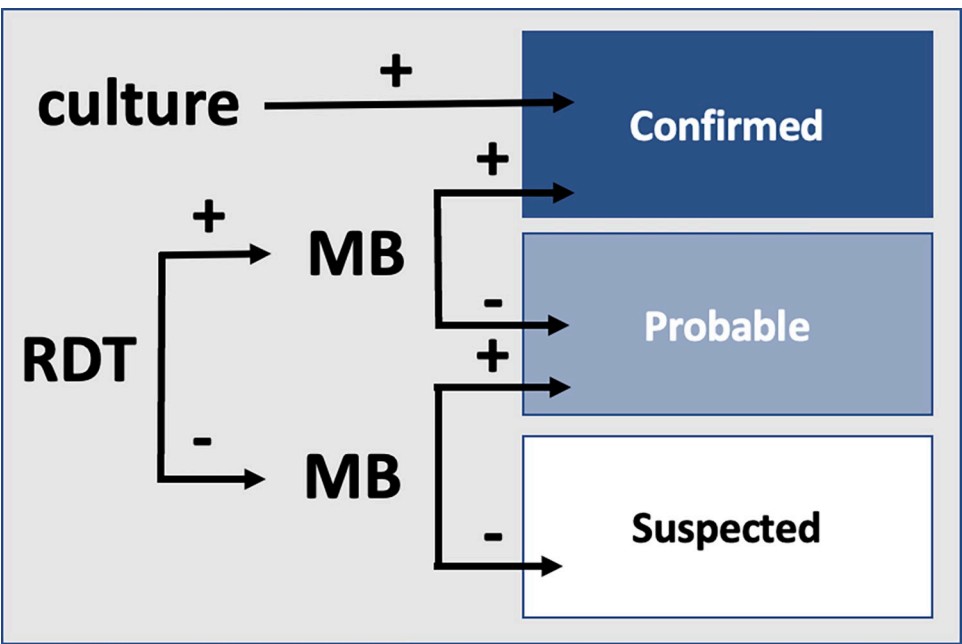

**Fig 2. Case classification algorithm.** Confirmed cases include cases with positive results for both RDT and MB and/or positive culture, probable have either RDT or MB positive, and suspected have no confirmatory laboratory results. MB, molecular biology; RDT, rapid diagnostic test.

specificity was around 80% for both PP (82%, 80% to 84%) and BP (85%, 81% to 89%) (Fig 3A and Table A in S1 Text). Additional analyses including an initially implemented classical polymerase chain reaction (cPCR) protocol confirm that these lacked specificity (PP: 55%, 52% to 58%, BP: 62%, 56% to 69%) (Table B in S1 Text) and justifies its timely replacement by MB. The latter was the most sensitive test (PP: 80%, 61% to 97%; BP: 95%, 86% to 100%), markedly higher than that of culture (PP: 7%, 0% to 23%; BP: 64%, 46% to 85%) and RDT (PP: 28%, 18% to 41%, BP: 72%, 61% to 83%) (Fig 3B and Table A in S1 Text). The statistical analysis also provides estimates of the performance of diagnostic tests that would be based on single gene diagnostic outcomes obtained from the quantitative PCR (qPCR) (Table A in S1 Text). Estimates were robust for deviations from model assumptions including the inclusion of the initial cPCR (Table B in S1 Text) and the use of a uniform prior on prevalence (Table D in S1 Text).

Under the assumption that samples were of good quality, we estimate that prevalence of infection among notified cases was 4% (3 to 7) for PP and 25% (18% to 28%) for BP (Fig 3C). This corresponds to 78 (50 to 119) and 81 (64 to 98) *Y. pestis* infections among notified PP (*N* = 1,779) and BP cases (*N* = 357), respectively. However, a challenge in diagnosing PP is the risk for samples to be of poor quality, i.e., that samples from a *Y. pestis*-infected individual do not contain detectable bacterial material. If a proportion of samples were of poor quality, estimates for the prevalence of infection would increase (Fig 3D). For example, in the extreme scenario where only 50% of samples were of good quality, estimates of the prevalence of infection would rise to 9% (6% to 13%) for PP and 45% (36% to 55%) for BP. For this analysis, we assumed sample quality to affect all tests equally. We also assessed a scenario in which test sensitivities were not fully independent and only the 2 qPCR gene results were affected by sample quality. This did not improve model fit (Fig B in S1 Text) and most parameter estimates were robust to departures from the assumption of test independence (Fig C in S1 Text).

We find that these estimates present good adequacy with the observed data [12] and can accurately reproduce (i) the number of notified cases classified as confirmed (PP: 19, 8 to 47

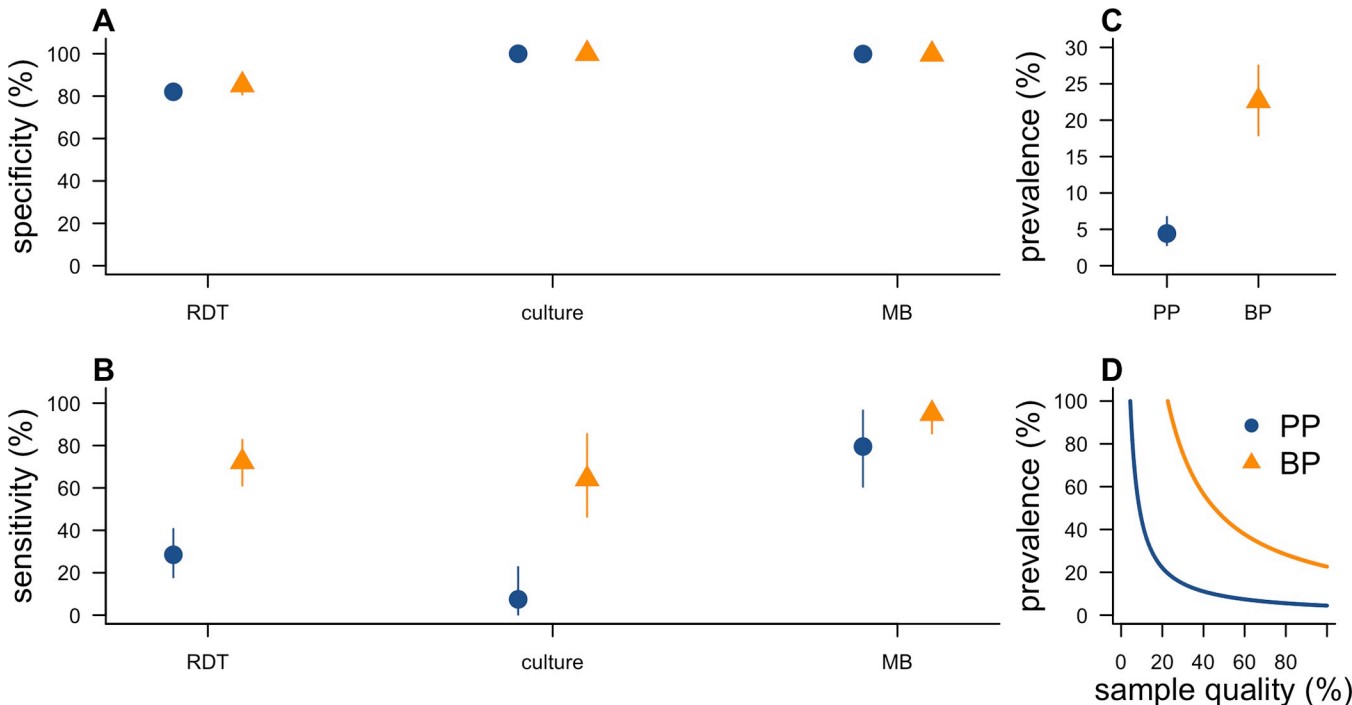

**Fig 3. Model estimates of test performance and prevalence.** (A) Specificity of each test, with RDT denoting rapid diagnostic test and MB denoting molecular biology. (B) Sensitivity of each test. (C) Prevalence of *Y. pestis* infection among notified cases, under the assumption of perfect sample quality. (D) Relationship between sample quality (i.e., the proportion of samples from infected individuals that contain detectable bacterial material) and estimated prevalence of infection among notified cases. Results are presented by clinical form: pneumonic (PP: blue) and bubonic (BP: orange). The circle/triangle shows the posterior median of the parameter while the lines show the 95% credible interval. The underlying data and code to reproduce this figure are available on Open Science Framework (https://osf.io/nbc4t/). BP, bubonic plague; MB, molecular biology; PP, pneumonic plague; RDT, rapid diagnostic test.

expected versus 27 observed; BP: 58, 37 to 81 versus 57) and probable (PP: 356, 338 to 377 versus 364; BP: 66, 45 to 87 versus 66) (Fig 1C); (ii) the number of notified cases testing positive for RDT, culture, or MB (Fig 1D); and (iii) the more detailed contingency table of the different diagnostic outcomes used for inference (Fig 1E).

Our analytical framework can be used to assess the performance of the case classification. For example, it can explain why the prevalence of *Y. pestis* among PP notified cases is estimated to be lower than the proportion of confirmed or probable cases (Fig 4A). In a scenario of low prevalence, the suboptimal specificity of RDT means that classification for PP based on confirmed or probable cases is characterized by a proportion of false positives (approx. 1-specificity) that is large relative to the prevalence. In contrast, a classification that solely relies on confirmed cases consistently underrepresents the prevalence due to low sensitivity of RDT and culture. For BP, the case classification performs well at any prevalence level, with the true prevalence always falling between the proportion of confirmed and confirmed/probable cases (Fig 4B and B panel of Fig D in S1 Text).

The positive predictive value (PPV) for a category of cases is the proportion of cases of that category that are *Y. pestis* infected. As expected, we find that the PPV of the confirmed or probable category is strongly impacted by prevalence among notified cases (Fig 4C and 4D). For example, if the prevalence of PP was 20%, over half of confirmed or probable cases would be expected to be *Y. pestis* infected. This proportion drops to as little as 22% (21% to 24%) for a prevalence of 5%. This shows that it is critical to avoid overreporting and ensure notified cases meet the clinical case definition. Cases classified as confirmed were, for both clinical forms,

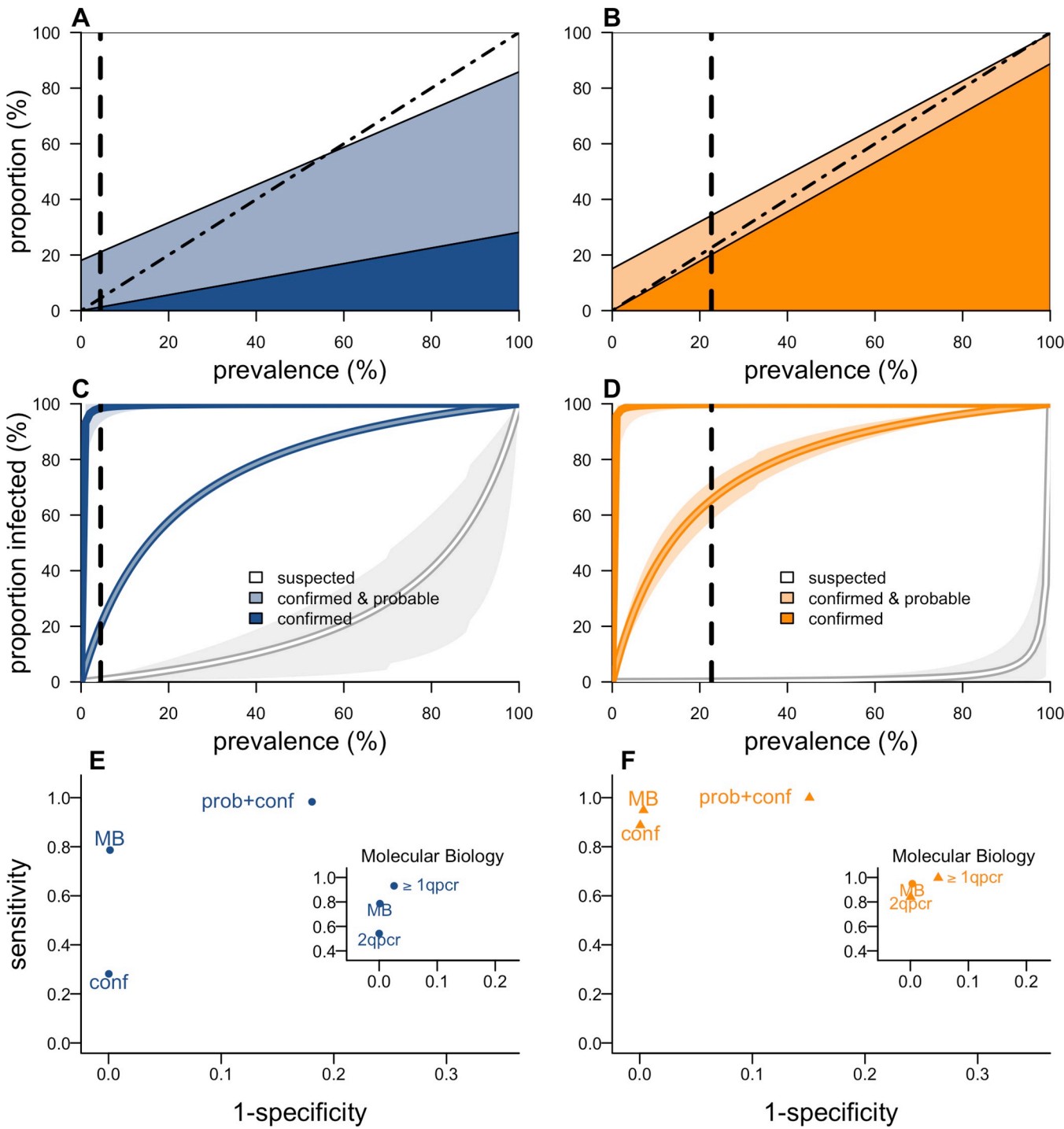

**Fig 4. Performance of the case classification system.** (A, B) Expected proportion of notified cases classified as confirmed (dark blue or orange), probable (light blue or orange), and suspected (white), as a function of prevalence of infection for PP (A) and BP (B). The dashed vertical line indicates the prevalence among notified cases estimated during the 2017 Madagascar outbreak. The dashed diagonal line corresponds to perfect classification (C, D). Expected proportion of *Y. pestis* infections among cases in the category confirmed, confirmed or probable, and suspected as a function of prevalence of infection for PP (C) and BP (D). (E, F) ROC plots presenting sensitivity versus (1-specificity) for a range of possible classification criteria for PP (E) and BP (F) and for simplifications of the MB algorithm for PP (inset of E) and BP (inset of F). MB is considered here due to its potential for being considered as a classifier by itself. Here, conf denotes confirmed and prob denotes probable. Classifications ≥1 qpcr and 2 qpcr represent results based on qPCR solely, i.e., in the absence of confirmatory cPCR, with ≥1 qpcr denoting "at least 1 gene positive" and 2 qpcr "both genes positive." The underlying data and code to reproduce this figure are available on Open Science Framework (https://osf.io/nbc4t/). BP, bubonic plague; cPCR, classical polymerase chain reaction; MB, molecular biology; PP, pneumonic plague; qPCR, quantitative polymerase chain reaction; ROC, Receiver operating characteristic.

almost all *Y. pestis* infected (PP: 98%, 91% to 100%; BP: 100%, 99% to 100%), deriving from perfect specificity of culture and the strict criterium requiring both RDT and MB to be positive. We further assess the risk of missing *Y. pestis*-infected cases and predict that 29% (16% to 42%) of *Y. pestis*-infected PP cases were classified as confirmed and 87% (73% to 98%) as confirmed or probable. This classification sensitivity is better for BP with 89% (81% to 96%) of infected cases being confirmed and 100% (99% to 100%) being confirmed or probable. The performance of case classification would be hampered if a substantial proportion of samples were of poor quality (Fig D in S1 Text).

We can also determine how to revise the classification system to minimize the proportions of false positive (1-specificity) and false negative cases (1-sensitivity) (Fig 4E and 4F). Best classification for both forms is based on MB, with a proportion of false positive and false negative cases, respectively, reduced from 2% to 0% (0% to 0%) and from 71% to 20% (3% to 40%) for PP (BP: 0% to 0%, 0% to 2% and 11% to 5%, 0% to 14%) (Fig 4E and 4F), providing a robust representation of the prevalence.

We then compare the MB algorithm (Fig A in S1 Text) to simpler alternatives that would not require confirmatory cPCR. We show that the MB algorithm is more sensitive than classification based on qPCR alone using "both genes positive" as a criterium and more specific than the one using "at least 1 gene positive" (Fig 4E and 4F).

Concordance between RDT and MB improved over time among negative MB samples (B and D panels of Fig E in S1 Text) but decreased among positive MB samples for PP (S5A Fig in S1 Text). We investigate possible changes in RDT performance during the epidemic. We find that RDT specificity increased significantly from 72% (69% to 76%) before week 41 to 95% (93% to 97%) afterward for PP (BP: 71%, 63% to 78% to 98%, 95% to 100%). Sensitivity of RDT was unchanged for BP (73%, 59% to 87% to 72%, 55% to 88%) but decreased for PP (34%, 16% to 53% to 14%, 3% to 30%) (Table C in S1 Text). Earlier and later cutoff times result in a lesser fit (Fig F in S1 Text). Estimates of RDT specificity for the second part of the outbreak are consistent with those obtained for the subsequent endemic BP season, during which the same batch was used (specificity: 99%, 96% to 100%), and are quite consistent with estimates from earlier evaluations of this test (64% sensitivity and 93% specificity based on latent class analysis) [11]. The 19% increase sensitivity estimated in the subsequent BP season (91%, 84% to 96%) suggests that outbreak-specific factors may have indeed hampered RDT and case classification performance in 2017 (Fig G in S1 Text).

Lastly, we can use our framework to derive, for each notified case, the probability of *Y. pestis* infection given their test results (i.e., the PPV). The probability is highest among cases with positive MB (100%) (Fig H in S1 Text) or culture (100%). We then use these estimates, together with the location and timing of cases, to reconstruct the dynamics of spread corrected for spatiotemporal variations in prevalence. Prevalence of *Y. pestis* infections among notified PP cases was 3-fold (BP: 2-fold) lower during the outbreak phase (weeks 39 to 43; when 75% of notifications occurred) than during the initial phase (Fig 5A and 5B). Such phenomenon is common when an outbreak receives a lot of attention from authorities, media, and communities, as was the case in 2017. Prevalence of *Y. pestis* infection among notified cases was highest in plague-endemic regions (BP: 3-fold higher than Antananarivo), where health personnel is accustomed to responding to BP (Fig 5C and 5D). Prevalence was lower among children (<5 year old) among notified BP cases, but not for PP (Fig 4E and 4F). Correcting for temporal variations in the prevalence, we find that the transmission of *Y. pestis* during this outbreak was less efficient than what was suggested by the analysis of notified cases, particularly for PP: The doubling time in the first 6 weeks was estimated to be 18 rather than 6 days (or 8 based on confirmed/probable) (BP: 24 versus 13 (17)) (Fig 5G and 5H).

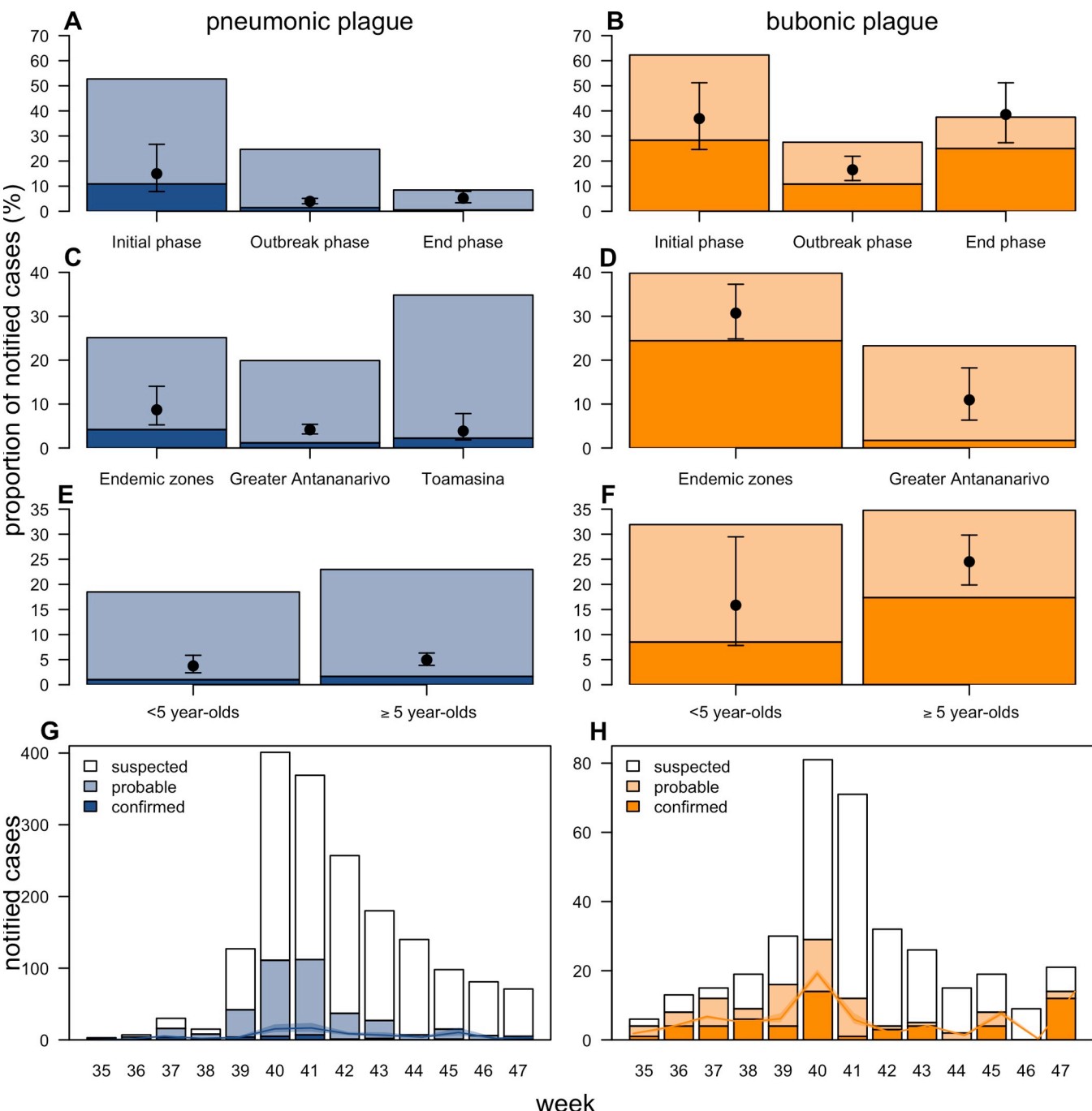

**Fig 5. Reconstruction of the outbreak by place and time.** (A, B) Estimated prevalence of infection among notified cases by time period for PP (A) and BP (B). Here, the initial phase spans weeks 34–38, outbreak phase 39–43, and the end phase 44–48. (C, D) Prevalence estimates by zone for PP (C) and BP (D). No BP cases were notified from Toamasina. (E, F) Prevalence estimates by age for PP (E) and BP (F). (G, H) Observed notifications (bars) vs. estimated infections (solid lines with shading denoting 95% credible intervals) among notified cases for PP (G) and BP (H). The stacked bar plots denote the percentage (A–F) and absolute numbers (G–H) by case classification. The underlying data and code to reproduce this figure are available on Open Science Framework (https://osf.io/nbc4t/). BP, bubonic plague; PP, pneumonic plague.

## Discussion

Assessing the true burden of an outbreak can be difficult in the absence of "gold standard" diagnostic. This can be especially problematic when scarce resources need to be allocated for

outbreak control. Here, using plague as a case study, we presented a statistical framework based on latent-class models to parse the results from multiple imperfect diagnostics and assess the true burden of the outbreak. We showed that around one tenth of notified cases were likely to be infected with *Y. pestis*. We showed that, particularly in scenarios with substantial misclassification of cases, poor specificity of some diagnostics can greatly skew case classification, even if a combination of diagnostic tests is used, and contribute to an overestimation of the true burden of infection. We used estimates of diagnostic test performance together with individual test results to reconstruct epidemiological trends in the proportion of true infections among notified cases and showed that misclassification of cases was highest during the peak of the epidemic and in regions nonendemic to plague. We illustrated the importance of optimized case classification algorithms, highlighting an overestimation of the transmission potential of the bacteria if based on the, typically used, tally of confirmed and probable cases.

This study highlights challenges inherent to plague diagnostics, particularly those of pneumonic cases. While specificity of test results was similar between bubon and sputum samples, the sensitivity of all diagnostics was substantially lower for sputum samples. This is in line with other respiratory illnesses such as pneumonia [10]. Poor-quality samples may well result in an underestimation of the true prevalence of infection among notified cases. We assessed how limited sample quality would affect our findings and showed that our general conclusion that the majority of notified cases were not infected with *Y. pestis* is robust to substantial amounts of poor-quality samples. We came to the same conclusion if sample quality issues affected some diagnostics more than others.

Classifying cases into confirmed, probable, and suspected is a routine public health effort that gives insight into the extent of the outbreak and an indication of the levels of uncertainty surrounding this. We highlight the importance of accurate classification algorithms and show that, particularly for diseases with nonspecific symptoms and high risks of misclassification (e.g., due to raised awareness or nonfamiliarity with the disease among public health responders), classification based on tests with poor specificity can result in vast overestimations of the outbreak extent. In the case of the plague outbreak in Madagascar, limited RDT specificity contributed to the majority of probable PP cases not to be infected with *Y. pestis*. We showed that the performance of the RDT improved toward the end of the outbreak. Such evolving RDT performance might be explained by the extreme circumstances surrounding this outbreak which may have resulted in changes within laboratories, e.g., due to overworked personnel or, conversely, changes in workflow to increase efficiency and proficiency of sample processing. It might also be due to a change of RDT batch that occurred in week 43. Assessing historical data [11] as well as data from a subsequent outbreak year indeed revealed better RDT test performance than was observed during the first half of the outbreak. The performance of case classification algorithms may therefore be better during nonoutbreak years and care should be taken to uphold this in crisis situations. Apart from upholding test performances, this may also include robust clinical case definitions to prevent large overreporting. Even under better conditions, however, the RDT is likely of limited value for case classification when other tools are available. Yet, RDTs are vital for point-of-care diagnostics in peripheral health settings, in particular for BP. Improving RDT performance should therefore be prioritized. Similarly, the inclusion of culture did not improve case classification, owing to its limited sensitivity. While the proportion of confirmed cases gave the best indication of the true proportion of infections, a large underestimation of the true burden of infections is expected in scenarios with less overreporting and a higher prevalence among notified cases. The inclusion of culture nevertheless remains fundamental for assessing circulating strains and the antibiotic resistance thereof. In this outbreak, this was particularly relevant as widespread use of prophylactic treatment was observed in response to the large volume of notified cases. The real risks

of resistance emergence associated with widespread use are another reason why accurate case classification is important.

MB had the best specificity and sensitivity. Especially for BP, adding other diagnostic tests to classify cases does not improve our ability to accurately classify cases. For PP, MB by itself would result in somewhat lower sensitivity than if used in combination with culture and RDT, but the reduced specificity outweighs this benefit. We also assessed whether the MB algorithm (Fig A in S1 Text) itself could be further improved. While the MB algorithm, particularly for PP, is somewhat less sensitive than the one using "at least 1 gene positive," we pose that increased specificity should be prioritized in low prevalence scenarios such as the one described here, confirming the relevance of the confirmatory cPCR performed in the MB algorithm.

The integrative framework presented here makes it possible to assess the performance of diagnostics, optimize their use for case classification, and reconstruct the whereabouts of infected cases during outbreaks in situations where no diagnostic gold standard is available. Improved case classification is particularly important for the allocation of scarce resources, for example, by accurately targeting contact tracing efforts and optimizing the impact of mobile test facilities. Beyond plague, such analytical framework could be a valuable tool to reduce uncertainties in other infectious disease outbreaks affected by nonoptimal diagnostics. This is particularly important when overreporting is likely due to nonspecific symptoms or if mass testing is applied. Above all, the development and availability of high-quality diagnostics remains a priority, particularly for pathogens prone to causing explosive outbreaks such as *Y. pestis*.

## Materials and methods

### Background information about plague in Madagascar

Madagascar accounts for 75% of plague cases worldwide [5]. Health professionals are required to notify all cases clinically suspected of *Y. pestis* infection to the CLP (WHO Collaborating Center) at the Institut Pasteur de Madagascar (IPM), where case notification forms are recorded and biological samples analyzed for laboratory confirmation. Treatment of cases is not contingent on biological confirmation of CLP. Annually, between 200 to 700, mostly bubonic (BP) (75%) cases are notified. The majority of these cases occur in the rural central highlands during the country's plague season (October to April). Occasional small outbreaks of PP were recorded in 1997, 2011, and 2015 in rural areas [15–18] and in 2004 in 1 commune of the county's capital city, Antananarivo.

Between August and November of 2017, the country experienced an outbreak that, with 2,414 notified cases, was much larger than regular plague seasons and presented with an unusual proportion of cases with clinically suspected PP.

### Data

Samples from clinically suspected cases (2,414) were sent to the CLP at the IPM for diagnostic testing. Treatment of suspected cases was not contingent on biological results. Biological samples were taken from cases presenting at health care settings with symptoms consistent with plague (i.e., for BP: presence of an isolated, painful adenopathy; for PP: cough (<5 days), bloody sputum, chest pain with fever) [3]. There was no formal clinical case definition that needed to be satisfied for patients to be tested. Samples included bubo aspirates from BP and secondary PP, sputum samples for PP, and liver and/or lung aspirates from deceased cases. All samples were tested for fraction 1 (F1) capsular antigen using an RDT [11]. Initially, MB was performed using cPCR targeting the *pla* gene on all samples. Due to low specificity of the cPCR, this test was abandoned on November 3 and replaced by real time qPCR targeting *pla* and *caf1* genes. If both genes tested positive, a sample was considered positive for MB. Samples with discordant or inconclusive qPCR results were verified using confirmatory cPCR on *pla*,

*caf1*, and *inv_1100* genes, with protocols improved to reach better specificity. They were considered positive upon positive results for the *inv_1100* gene and/or for both *pla* and *caf1* genes (see Fig A in S1 Text for decision tree) [19]. All samples received before November 3 were retested using the MB protocols (November to December 2017). In addition, culture was performed on all samples with positive RDT [20]. PP samples from nonendemic regions received between September 11 and October 3 were cultured irrespective of RDT result. No serological testing was performed during the outbreak.

As per WHO guidelines, cases were classified based on their diagnostic test results as *confirmed* if culture and/or both RDT and MB were positive, *probable* upon positive results for either MB or RDT, and *suspected* otherwise (Fig 2). Initial cPCR results were not considered for case classification. Culture is often regarded as a gold standard given its perfect specificity yet lacks sensitivity. Culture sensitivity may have been particularly challenged during this outbreak as a result of widespread prophylactic antibiotic use [9].

## Estimating diagnostic test performances and burden of infection

We develop a statistical model based on latent class methods to estimate test performances and burden of infection among the population of notified cases (*N*). Here, we distinguish diagnostic outcomes as the raw outcomes of performed qPCRs (i.e., gene-specific outcomes) and the composite result of the confirmatory cPCR (Fig A in S1 Text). The composite of these makes up the result of a diagnostic test. For each notified case (*i*) dichotomous results ($y_{ij}$) are available for up to *J* diagnostic outcomes (*j*) (i.e., 1 for each of RDT and culture and up to 3 for MB: 2 for qPCR (*pla*, *caf1*) and 1 for confirmatory cPCR (*pla*, *caf1*, *inv_1100*), with $y_{ij} = 1$ denoting a positive and $y_{ij} = 0$, a negative result. The infection status of case *i* is denoted $d_i$ (= 1 if infected and 0 otherwise). The sensitivity and specificity of diagnostic outcome *j* are denoted $S_j = P(Y_j = 1|D = 1)$ and $C_j = P(Y_j = 0|D = 0)$, respectively.

Here, we calculate the contribution to the likelihood of the different diagnostic outcomes. Test-specific sensitivities and specificities are then calculated from the characteristics of the diagnostic outcomes that make up a specific test (MB in particular). We first discuss the likelihood for those diagnostic outcomes that are performed irrespective of other diagnostic outcomes (RDT, qPCR), followed by those that are performed conditional on other diagnostic outcomes (culture and confirmatory cPCR).

**Contribution to the likelihood of RDT, qPCR (pla, caf1).** We first calculate the contribution to the likelihood of diagnostic outcomes that are performed irrespective of other diagnostic outcomes, namely RDT, qPCR (*pla* and *caf1*) (indexed in Eq 1 as 1...*U*). If the infection status of a case was known, 2 conditional probabilities would have to be considered:

- Conditional on being infected by plague and given model parameters *θ*, the joint probability of test results for case *i* is

$$P(\{y_{ij}\}_{j=1...U}|\theta, d_i = 1) = \prod_{j=1}^{U} S_j^{y_{ij}}(1 - S_j)^{1-y_{ij}}. \tag{1}$$

- Conditional on not being infected by plague and given model parameters *θ*, this probability becomes:

$$P(\{y_{ij}\}_{j=1...U}|\theta, d_i = 0) = \prod_{j=1}^{U} C_j^{1-y_{ij}}(1 - C_j)^{y_{ij}}. \tag{2}$$

In practice, the infection status of a case is unknown, and we therefore work on the

unconditional joint probability of diagnostic outcomes for case $i$ that integrates over the different possibilities:

$$P(\{y_{ij}\}_{j=1...U}|\theta) = P(d_i = 1)P(\{y_{ij}\}_{j=1...U}|\theta, d_i = 1) + P(d_i = 0)P(\{y_{ij}\}_{j=1...U}|\theta, d_i = 0)$$

$$P(\{y_{ij}\}_{j=1...U}|\theta) = \pi \prod_{j=1}^{U} S_j^{y_{ij}}(1 - S_j)^{1-y_{ij}} + (1 - \pi) \prod_{j=1}^{U} C_j^{1-y_{ij}}(1 - C_j)^{y_{ij}} \tag{3}$$

where $\pi$ is the prevalence of plague infection among notified cases.

While RDT and qPCR (*pla* and *caf1*) were performed independent of other results, culture was done based on RDT outcome, period, and zone, and confirmatory cPCR (*pla*, *caf1*, $inv_{1100}$) was performed only if qPCR was inconclusive. We need to integrate such conditioning in our analysis to avoid biases.

**Contribution to the likelihood of culture.** For PP samples received from a nonendemic region between September 11 and October 3, culture was performed irrespective of RDT results and we therefore use the formulation described above. For all other samples, culture was performed only if RDT was positive. Hence, the conditional probability for these individuals to obtain a culture result ($y_{i\ cult}$) is

$$P\big(y_{i\ cult}|y_{i\ RDT} = 1, \theta\big)$$
$$= \pi \frac{S_{cult}^{y_{cult}}(1 - S_{cult})^{1-y_{cult}} S_{RDT}}{P(y_{iRDT} = 1)} + (1 - \pi)\frac{C_{cult}^{1-y_{cult}}(1 - C_{cult})^{y_{cult}}(1 - C_{RDT})}{P(y_{iRDT} = 1)}, \tag{4}$$

which can also be expressed in terms of the PPV (the proportion infected among individuals with a positive test result) of RDT

$$PPV_{RDT} S_{cult}^{y_{cult}}(1 - S_{cult})^{1-y_{cult}} + (1 - PPV_{RDT})C_{cult}^{1-y_{cult}}(1 - C_{cult})^{y_{cult}}, \tag{5}$$

where PPV is

$$PPV_{RDT} = \frac{\pi S_{RDT}}{\pi S_{RDT} + (1 - \pi)(1 - C_{RDT})}. \tag{6}$$

Results from culture that did not adhere to this conditioning (PP: 283, BP: 60) were not included in the analysis as the reason for this additional testing cannot be traced back but was likely nonrandom and affected by other test results.

**Contribution to the likelihood of cPCR (pla, caf1, and inv$_{1100}$).** Test results for MB were a composite of up to 5 diagnostic outcomes (Fig 2). Confirmatory cPCR for genes 1 to $k$ were performed conditional on discordant qPCR results. The composite result of this test was included in the model.

$$P\big(y_{i confPCR,}|Y_{i\ qPCR_{pla}} \neq Y_{i\ qPCR_{caf1}}, \theta\big) =$$

$$\pi \frac{S_{confPCR}^{y_{confPCR}}(1 - S_{confPCR})^{1-y_{confPCR}} * (S_{pla} * (1 - S_{caf1}) + (1 - S_{pla}) * S_{caf1})}{P(Y_{i\ qPCR_{pla}} \neq Y_{i\ qPCR_{caf1}})} +$$

$$(1 - \pi)\frac{C_{confPCR}^{1-y_{confPCR}}(1 - C_{confPCR})^{y_{confPCR}} * (C_{pla} * (1 - C_{caf1}) + (1 - C_{pla}) * C_{caf1})}{P(Y_{i\ qPCR_{pla}} \neq Y_{i\ qPCR_{caf1}})}. \tag{7}$$

**Joint likelihood.** The likelihood per case is the product of terms described in Eqs 3, 4, and 7:

$$L_i = \prod_i P(\{y_{ij}\}_{j=1\ldots U}|\theta) * \prod_{i:y_{i\ RDT}=1} P(y_{i\ cult}y_{i\ RDT} = 1)*$$

$$\prod_{i:Y_{i\ qPCR_{pla}} \neq Y_{i\ qPCR_{caf1}}} P(y_{iconfPCR_{pla}}, Y_{i\ qPCR_{pla}} \neq Y_{i\ qPCR_{caf1}}). \tag{8}$$

**Inference.** Parameter estimation was done in a Bayesian setting using a Bayesian Metropolis–Hastings MCMC approach [14]. We utilized a weakly informative beta-distributed prior for prevalence (shape = 1, scale = 2) (i.e., chance of prevalence being below 50% is twice as high as above) based on estimates of prevalence from previous BP outbreaks [21]. To confirm the robustness of the results to the choice of priors on prevalence, the MCMC was also performed with a uniform prior between 0 and 1 (Table D in S1 Text). For specificities of tests associated with MB, beta-distributed priors were used with means of 95% (shape = 12.7, scale = 0.67) based on verifications done in the IPM laboratories prior to implementation. The specificity of culture was fixed at 100%. For all other parameters, we used uniform priors between 0 and 1 (i.e., for all sensitivities as well as the specificity of RDT). Metropolis–Hastings updates were performed on a natural scale with step sizes adjusted such to obtain an acceptance probability between 10% and 50% [14]. Traces of the MCMC were plotted per parameter and convergence was assessed visually (Figs I and J in S1 Text).

**Assuming imperfect sample quality.** Collection of good-quality samples is challenging and might be affected by prophylactic treatment, preservation techniques, and the delays between symptom onset and sample testing at CLP. The prevalence of infection is related to the prevalence of detectable bacterial material in the collected samples ($\tau$) such that $\tau = \rho\pi$, where $\rho$ is the probability of good sample quality given a truly infected individual. Accounting for $\rho$ in Eq 3 gives

$$P(\{y_{ij}\}_{j=1\ldots J}) =$$

$$\pi \prod_{j=1}^{J}(\rho(S_j^{y_j}(1 - S_j)^{1-y_j}) + (1 - \rho)C_j^{1-y_j}(1 - C_j)^{y_j})+$$

$$(1 - \pi)\prod_{j=1}^{J} C_j^{1-y_j}(1 - C_j)^{y_j}. \tag{9}$$

Here, $S_j$ and $C_j$ denote the absolute sensitivity and specificity, i.e., assuming the sample is of good quality. The definition of $\rho$ implies that all tests are equally affected by factors reducing sample quality.

**Assuming dependence between qPCR results.** While in the above calculations, test results are assumed independent of each other, in practice this may not always be true. Notably, as results from both genes assessed by qPCR are performed in the same assay, possible contaminations or technical problems might affect both test outcomes concurrently. To assess the sensitivity of our results to departures from the assumption of independence, we adjusted the contribution of the outcomes qPCR$_{pla}$ and qPCR$_{caf}$ to the likelihood (Eq 8) to reflect a larger likelihood of concordance between both diagnostic outcomes

$$P(\{y_1, y_{2,}\}|\theta, d_i = 1) = S_1^{y_1} * S_2^{y_2}(1 - S_1)^{1-y_1}(1 - S_2)^{1-y_2} + (-1)^{y_1-y_2}(cov_{12}|d = 1). \tag{10}$$

Here, indices 1 and 2 refer to qPCR *pla* and *caf*, respectively, and *cov* denotes the pairwise covariance between diagnostic outcomes [22]. We assess whether the inclusion of a covariance factor affected the fit to the data, using DIC as an indicator of fit [23], and whether estimated test characteristics were robust to departures from assumptions of independence.

Model fit did not improve upon the inclusion of a covariance factor (Fig B in S1 Text) and parameter estimates were relatively robust: Prevalence of PP was insensitive to the existence of correlations between these tests (4% versus 6%) (Fig C in S1 Text). Prevalence estimates of BP increased for high levels of correlation (35% versus 23%), which came with increased RDT specificity (97% versus 85%) and reduced sensitivity of culture (41% versus 64%).

## Detecting heterogeneity in test performance

**Over the course of the outbreak.** Changes in concordance between RDT and MB were observed toward the end of the outbreak (Fig E in S1 Text). To test whether a change in RDT-test performance could explain this observation, we reran the inference routine allowing RDT to have different test characteristics before and after a predetermined cutoff week (i.e., essentially treating RDT before and after the cutoff point as distinct tests). We used different cutoff weeks (38 to 43), where the week denotes, as elsewhere in the manuscript, the date of symptom onset of the cases. The model with the cutoff week that yielded the highest likelihood was then compared to the baseline model (i.e., assuming no change in RDT performance) using DIC to test whether the change in RDT performance resulted in an improved model fit. We did not examine changes in other tests because (i) cPCR was terminated halfway through the outbreak; (ii) culture only yielded few positive samples; and (iii) MB was performed (in retrospect) at the end of the outbreak.

**Between outbreaks.** We compared results from the outbreak year (2017 to 2018) with those from the subsequent plague season (2018 to 2019). Between August 17, 2018 and April 7, 2019, 261 (46 PP, 211 BP, 4 unknown form) cases were reported to the CLP. All samples were analyzed using MB, RDT, and culture, with the same protocols as were used during the 2017 outbreak year. Among sputum samples for PP ($n = 25$), 22 were negative for both MB and RDT. Others were positive for either or both tests (1 MB+ and RDT+, 1 MB− and RDT+, 1 MB+ and RDT−), with those positive for MB confirmed by culture. Among bubon aspirates for BP ($n = 194$), 174 were concordant between MB and RDT (94 positive, 80 negative). Given the low number of positive sputum samples for PP (1 confirmed, 2 probable, 22 suspected), we only analyzed BP samples from this season. Inference was similar to that on samples from 2017, but since culture was performed on all samples, no conditioning was needed when assessing the contribution of culture on the joint likelihood. In addition, due to high concordance between both genes used for qPCR, few conditional PCRs were performed. We thus did not estimate the performance of conditional PCRs during this season.

## Outbreak reconstruction

We derived the probability of *Y. pestis* infection for each notified case based on the PPV associated with their results and assuming the medians of the estimated prevalence, sensitivity, and specificity (see Eq 6). The sum of all PPVs denotes the expected number of true infections among notified cases. We used this relationship to reconstruct the number of expected infections by subgroup. We divided the notified cases according to the following categories: (i) by period, distinguishing the initial phase (weeks 34 to 38), the outbreak phase (weeks 39 to 43), and the end phase (weeks 44 to 48); (ii) by week; (iii) by zone, distinguishing endemic zones (plague-endemic districts [24] apart from greater Antananarivo), greater Antananarivo (urban community of Antananarivo and the 3 neighboring districts), and Toamasina district; and (iv)

by age group (below and above 5 years of age). Using these, we estimated the prevalence and exact binomial 95% confidence interval of infection among notified cases by subgroup.

## Software

All analyses have been performed in R [25]. MCMC results have been processed using the *coda* [26] and *BayesianTools* packages [27].

## Supporting information

**S1 Text. Supplementary information appendix.** Fig A. Molecular biology (MB) algorithm. Fig B. Model fit as a function of covariance between qPCR$_{pla}$ and qPCR$_{caf1}$ sensitivities for pneumonic forms (A) and bubonic forms (B). Fig C. Sensitivity of parameter estimates to different levels of correlation between the sensitivity of qPCR$_{pla}$ and qPCR$_{caf1}$. Fig D. Performance of case classification system assuming sample quality of 75%. Fig E. RDT vs. MB concordance over time. Fig F. Model fit as a function of the timing of changed RDT performance for pneumonic (PP) (A) and bubonic plague (BP) (B). Fig G. ROC plots presenting for a range of possible classification criteria for pneumonic (PP) (A, C) and bubonic plague (BP) (B, D, E) before (A, B) and after week 41 (C, D) and during the 2018 endemic season (E). Fig H. Distribution of positive predictive values (PPVs) by test result and clinical form. Fig I. Traceplots for MCMC of default model for pneumonic forms. Fig J. Traceplots for MCMC of default model for bubonic forms. Table A. Model estimates of the performance of RDT, culture, MB, and of tests that would be based on single diagnostic outcomes. Table B. Model estimates of test performance of RDT, culture, MB, and of tests that would be based on single diagnostic outcomes. In addition to the default analysis presented in Table A in S1 Text, here, the initial cPCR was included in the analysis. Results of this test were removed from the final analysis because performances of that test were too low. The results of the initial cPCR were not considered in the case classification. Table C. Model estimates of the performance of RDT, culture, MB, and of tests that would be based on single diagnostic outcomes, in a scenario change in RDT performance at week 41 of the outbreak. Table D. Model estimates of the performance of RDT, culture, MB, and of tests that would be based on single diagnostic outcomes, with a noninformative uniform prior on the prevalence of infection among notified cases. (DOCX)

## Acknowledgments

The authors would like to thank for their continuous support during the epidemic the other team members of the IPM laboratories (from the Virology, Experimental Bacteriology, Infectious Diseases Immunology, Malaria Research Units, the Clinical Biology Laboratory, and the Hygiene Food and Environment Laboratory), members of supporting research units at IP Paris (CIBU, Epidemiology of Emerging Infectious Diseases Unit, Molecular Genetics of RNA Viruses Unit), experts deployed through the Global Outbreak Alert Response Network (GOARN), and all the colleagues who were instrumental in improving and implementing epidemiological and laboratory surveillance and supported the epidemic response efforts.

## Author Contributions

**Conceptualization:** Quirine ten Bosch, Birgit Nikolay, Juliette Paireau, Laurence Baril, Minoarisoa Rajerison, Simon Cauchemez.

**Data curation:** Voahangy Andrianaivoarimanana, Soloandry Rahajandraibe, Rindra Randremanana, Laurence Baril, Minoarisoa Rajerison.

**Formal analysis:** Quirine ten Bosch, Simon Cauchemez.

**Investigation:** Voahangy Andrianaivoarimanana, Beza Ramasindrazana, Guillain Mikaty, Rado J. L. Rakotonanahary, Soloandry Rahajandraibe, Maxence Feher, Quentin Grassin, Soanandrasana Rahelinirina, Feno Rakotoarimanana, Marie Melocco, Javier Pizarro-Cerdá, Anne-Sophie Le Guern, Minoarisoa Rajerison.

**Methodology:** Quirine ten Bosch, Simon Cauchemez.

**Project administration:** Voahangy Andrianaivoarimanana, Voahangy Rasolofo, Eric Bertherat, Maherisoa Ratsitorahina, André Spiegel, Laurence Baril, Minoarisoa Rajerison.

**Writing – original draft:** Quirine ten Bosch, Voahangy Andrianaivoarimanana, Beza Ramasindrazana, André Spiegel, Laurence Baril, Minoarisoa Rajerison, Simon Cauchemez.

**Writing – review & editing:** Guillain Mikaty, Rado J. L. Rakotonanahary, Birgit Nikolay, Soloandry Rahajandraibe, Maxence Feher, Quentin Grassin, Juliette Paireau, Soanandrasana Rahelinirina, Rindra Randremanana, Feno Rakotoarimanana, Marie Melocco, Voahangy Rasolofo, Javier Pizarro-Cerdá, Anne-Sophie Le Guern, Eric Bertherat, Maherisoa Ratsitorahina.

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
