## [Editor Report · Decision Letter 0]

3 Dec 2021

Dear Dr. Ten Bosch, 

Thank you for submitting your manuscript entitled "Evaluating and optimizing the use of diagnostics during epidemics: Application to the 2017 plague outbreak in Madagascar" for consideration as a Research Article by PLOS Biology. I apologize for the time you have been waiting for a decision while we were discussing about your manuscript.

Your manuscript has now been evaluated by the PLOS Biology editorial staff, as well as by an academic editor with relevant expertise, and I am writing to let you know that we would like to send your submission out for external peer review.

Once your full submission is complete, your paper will undergo a series of checks in preparation for peer review. Once your manuscript has passed the checks it will be sent out for review. To provide the metadata for your submission, please Login to Editorial Manager (https://www.editorialmanager.com/pbiology) within two working days, i.e. by Dec 05 2021 11:59PM.

If your manuscript has been previously reviewed at another journal, PLOS Biology is willing to work with those reviews in order to avoid re-starting the process. Submission of the previous reviews is entirely optional and our ability to use them effectively will depend on the willingness of the previous journal to confirm the content of the reports and share the reviewer identities. Please note that we reserve the right to invite additional reviewers if we consider that additional/independent reviewers are needed, although we aim to avoid this as far as possible. In our experience, working with previous reviews does save time. 

If you would like to send previous reviewer reports to us, please email me at pjaureguionieva@plos.org to let me know, including the name of the previous journal and the manuscript ID the study was given, as well as attaching a point-by-point response to reviewers that details how you have or plan to address the reviewers' concerns. 

Given the disruptions resulting from the ongoing COVID-19 pandemic, please expect some delays in the editorial process. We apologise in advance for any inconvenience caused and will do our best to minimize impact as far as possible.

Kind regards,

Paula

Paula Jauregui, PhD

Editor

PLOS Biology

---

## [Decision Letter · Decision Letter 1]

27 Feb 2022

Dear Dr Ten Bosch,

Thank you for submitting your manuscript "Evaluating and optimizing the use of diagnostics during epidemics: Application to the 2017 plague outbreak in Madagascar" for consideration as a Research Article at PLOS Biology. Your manuscript has been evaluated by the PLOS Biology editors, an Academic Editor with relevant expertise, and by several independent reviewers.

As you will see in the reviews pasted below, all reviewers appreciate the importance and accuracy of the work, and raise several points about methodology reporting and presentation/interpretation of the results. Please pay close attention to the technical concerns raised by Reviewer 2, and to ensuring that the findings are robust once reviewer reviewer 2's concerns are all clarified or corrected. 

In light of the reviews, we are pleased to offer you the opportunity to address the comments from the reviewers in a revised version that we anticipate should not take you very long. We will then assess your revised manuscript and your response to the reviewers' comments and we may consult the reviewers and the Academic Editor again. We also request that your please address the following data and other policy-related requests:

1) Data: you may be aware of the PLOS Data Policy, which requires that all data be made available without restriction: http://journals.plos.org/plosbiology/s/data-availability. For more information, please also see this editorial: http://dx.doi.org/10.1371/journal.pbio.1001797

Note that we do not require all raw data. Rather, we ask for all individual quantitative observations that underlie the data summarized in the figures and results of your paper. For an example see here: http://www.plosbiology.org/article/info%3Adoi%2F10.1371%2Fjournal.pbio.1001908#s5

These data can be made available in one of the following forms:

I) Supplementary files (e.g., excel). Please ensure that all data files are uploaded as 'Supporting Information' and are invariably referred to (in the manuscript, figure legends, and the Description field when uploading your files) using the following format verbatim: S1 Data, S2 Data, etc. Multiple panels of a single or even several figures can be included as multiple sheets in one excel file that is saved using exactly the following convention: S1_Data.xlsx (using an underscore).

II) Deposition in a publicly available repository. Please also provide the accession code or a reviewer link so that we may view your data before publication.

Regardless of the method selected, please ensure that you provide the individual numerical values that underlie the summary data displayed in the following figure panels: Figures 1 A–E , 2 A–D, 3 A–F, 4 A–H, S 3 AB, S 4, S 5 A–D, S 6 A–D, S 7 AB, S 8 A–E, S 9 , Table S1, Table S 2, Table S3.

1.1) Please also ensure that each figure legend in your manuscript includes information on where the underlying data can be found and that your supplemental data file/s has/have a legend.

1.2) Please ensure that your Data Statement in the submission system accurately describes where your data can be found.

We expect to receive your revised manuscript within 1 month.

**IMPORTANT - SUBMITTING YOUR REVISION**

Your revisions should address the specific points made by each reviewer, as well our editorial requests outlined above. Please submit the following files along with your revised manuscript:

*Resubmission Checklist*

*Published Peer Review*

Sincerely,

Dario

Dario Ummarino, PhD

Senior Editor

PLOS Biology

dummarino@plos.org

REVIEWS:

Reviewer's reports:

Reviewer #1: This manuscript employs Bayesian latent class analysis to better estimate the test characteristics of diagnostic tests used during an outbreak of plague in Madagascar, as well as to estimate the true prevalence of plague among reported cases. This type of analysis is increasingly being used to better understand medical diagnostics, particularly in cases for which no clear gold standard exists. The authors' work is novel; while the methods used are well-described they have not previously been applied to plague diagnostics. The manuscript is well-written and clear. The methods are also well-described; specifically the authors clearly delineate the prior distributions used for each parameter as well as the type of covariance structure used to evaluate the assumption of conditional independence. While one might have chosen different methods to model conditional dependence (e.g. the random effect model of Qu et al.), the authors' choice is reasonable and appropriate for this purpose. 

Minor comments:

- The authors did not mention what diagnostics were used to assess MCMC convergence; this should be explicitly stated in the methods.

- The title for Table S2 would benefit from explicitly stating that this table was based on a model in which the prior for specificity was a uniform beta distribution (this is suggested in the text)

-Several of the supplemental figures were not referenced/mentioned in the text. Most of these were fairly well-explained, but it was not clear what Figure S9 was demonstrating. It would be helpful to have some mention of the supplemental figures either in the main text or perhaps in expanded/more descriptive figure legends (starting on line 491).

-The authors state that they will make their code available on a specified github site. It would have been helpful for the review to have included the code as a supplemental file.

Reviewer #2: The authors set out to address a major challenge in epidemiology: uncovering the spread of a pathogen when only imperfect diagnostic tests are available. I found the study to be well designed and enjoyed reading the manuscript. The approach used in this study has potential to be of broad interest for epidemiologists studying wide range of infectious disease. However, I do have some concerns, in particular with: i) the scope of the results, ii) the clarity/comprehensiveness of the Methods, and iii) the clarity of some figures. Provided these concerns can be addressed, I expect the manuscript to be suitable for publication in PLoS Biology.

Specific concerns:

Scope of results:

1. The estimates of prevalence of both BP and PP among notified cases seem very similar to their respective fractions of notified cases that are confirmed (i.e. confirmed using the imperfect diagnostic tests). I feel the study's importance would be much more evident if the deviation between estimated prevalence and fraction confirmed was clearly demonstrated, and its consequences explored. The authors state that "We showed that around one tenth of notified cases were likely to be infected with Yersinia pestis", but to my eye this looks comparable to the fraction of confirmed cases in the data (Figure 1C)?

2. The author's results suggest that both "suspected" and "probable cases" are extremely unreliable indicators for whether an individual is infected with plague (with suspected being far worse). Do you know why both are so bad? Also could you state the diagnostic criteria used to determine whether someone is suspected vs. probable? I'm not an expect on plague diagnosis, but I would imagine some of the symptoms that medical practitioners are using to inform their diagnosis (of bubonic plague in particular) are pretty unique, so I'm surprised they are doing so poorly.

3.The poor sensitivity of both RDT and culture suggest there would be a massive underestimation of PP cases if they were used as primary diagnostic methods. Could you explain why this underestimation is not the case (I'm trying to compare Figure 1A with Figure 4G taking into account the results in Figure 2).

4. To what extent do the sensitivities and specificities shown in Figure 2 deviate from what was known before the study? If the tests come with stated sensitivities and specificities (which may, in light of this study, be inaccurate) it would helpful to state them here.

Likelihood function and data:

5. Estimating diagnostic test performance. I found the derivation and explanation of the per case likelihood function to be well reasoned and mostly clear, but a few changes would improve clarity. 

Firstly, on lines 302-305 the diagnostic outcomes are indexed "1,…U" and then introduced explicitly "(i.e. RDT, qPCR (pla and caf1)". The relationship between the different test listed and the indexing (j = 1,…,U) took a bit of time to figure out. I suggest flipping the order, i.e. introducing the four tests performed irrespective of other outcomes and then explain how they are indexed. Also, if U = 4 then explicitly state this (or leave out the parameter U and replace it with 4 throughout). Finally, and this may be a question of personal preference, but when I see "y_ij" my mind thinks elements of a square matrix (e.g. correlation or distance matrices). Given the two indices label different things, I suggest changing one to a non-neighbouring letter (e.g. a or μ)

6. Eq. 8 is the likelihood per case. I suggest adding a subscript to the left-hand side to indicate this (e.g.. L_i).

7. The main text and figures suggest that various different disaggregations of the data were performed (e.g. by spatial location or time of sampling). I found which data were used in which fit a bit challenging. It would be helpful to summarise/list these in a section of the methods, and explicitly state what was done in each scenario (e.g. something along the lines "the MCMC inference procedure was re-ran for each dataset individually").

MCMC:

8. The authors should explicitly list of all parameters estimated (perhaps as a table with the priors also in a column). At present the methods state "[w]e used uniform priors between 0 and 1 for most parameters", but I am unclear what those parameters are.

9. The authors should state the exact MCMC algorithm used (e.g. "the Metropolis-Hastings algorithm with No-U-Turn sampler"). Additionally the authors should state the implementation of the algorithm (e.g. "implemented in python package pymc3"). 

10. I'm a bit concerned by the use of a weakly-informative beta-distributed prior for prevalence, given it is "based on estimates of prevalence from previous BP outbreaks". One of the findings of this study is that estimates of confirmed BP prevalence can be wrong -- so what's stopping this previous study suffering from the same diagnostic issues as present in the 2017 plague season. Is it necessary to assume this prior distribution and how does it impact on the study findings?

Figure 1

11. I was unclear exactly what the proportions in Figure 1D and E correspond to. Are they the number of samples with a positive test result divided by the number of notified cases? If so, eyeballing panel B suggests that ~40% is too high for bubonic plague and culture, even if all confirmed cases are confirmed via culture. Or is it the proportion of culture samples that were positive? I think the caption needs revising to make clear exactly what is being shown. Furthermore, given the y-axis ranges aren't that different between panels C-E, I suggest making them all the same.

12. Are the "confidence intervals" really confidence intervals? The approach here is Bayesian — how are the confidence intervals calculated?

13. I wonder if panel A and B might benefit from being plotted on a log or square root scale.

Figure 2

14. Figure 2 would be improved by labelling all y-axis. 

Figure 3

14. I think clarity would be improved by changing the caption to read prevalence of infection amongst notified cases (consistent with rest of paper). 

15. I take it the dashed diagonal line in A and B corresponds to a perfect classifier (sensitivity = specificity = 1). It would be helpful to mention this in the caption. 

16. I enjoyed the ROC curve plots and found them very informative. However, I was unsure what "conf" and "prob" meant in this context (is it the same as confirmed and probable above?) and how their sensitivity and specificity were calculated. Furthermore, why was only molecular biology plotted here? Would't it be informative to also show the other diagnostic methods?

Figure 4

17. Are these prevalence estimates among notified cases or overall population prevalence estimates (including cases that were not notified)?

18. It would be helpful to compare each panel to the respective data from each grouping. E.g. total fraction of notified cases that are confirmed/suspected.

19. The multi-y-axis in panels G and H fooled me for a long time. I spent a while confused how estimated infection among notified could be larger than notified. I suggest the authors plot both time series on the same y scale, maybe using a log or square root scale. If the authors want to stick with having a twinned y axis then the right-hand scale also needs labelling to make it clearer. Furthermore, I really think it would be helpful to also show the data on confirmed and suspected cases in these panels — i.e. what is shown in Figure 1A and B.

Reviewer #3: This is a post-hoc analysis of an outbreak of plague caused by the bacterium Yersinia pestis in Madagascar in 2017 to 2018. The goal is to more accurately determine the size, timing, and spatial extent of the outbreak. The issue is that testing criteria for properly diagnosing plague generally have low sensitivity, especially for detection based on culturing of bacteria and a rapid diagnostic test based (RDT) on detecting antigens to proteins of the F1 capsule. Tests based on molecular biology (MB) use qPCR to detect the pla and caf1 genes have higher sensitivity. The problem is that the true gold standard, culturing of bacteria, has especially low sensitivity for sputum samples for pneumonic plague, and only slightly higher sensitivity for samples aspirated from lymph nodes for bubonic plague. A final MB qPCR test for a third gene inv1100 is highly sensitive and makes final confirmation more secure.

The main conclusion is that the best estimates for prevalence of confirmed and probable cases among suspected cases was only 5% for pneumonic plague, and 25% for bubonic plague. Thus, the outbreaks were not as large as initially suspected and reported. Further, neither culturing nor RDT provided reliable information. Higher accuracy in the case of both tests could have affected public health responses to the outbreaks. qPCR tests, especially for all three gene fragments, provide the most reliable testing. Nonetheless, the problem is still whether the testing would have provided actionable decisions, based both on how quickly qPCR tests can be turned around, and because prophylactic antibiotic treatment during outbreaks likely both suppressed the outbreak and affected the qPCR test sensitivity. 

The study nonetheless provides a good framework for analysis of outbreaks, if the sensitivity and specificity of the tests are known. 

One aspect that is not discussed at all is whether effective testing could lead to more judicious use of antibiotics, and potentially help slow antibiotic resistance by the plague pathogen. Keeping off the antibiotic treadmill as one antibiotic after another loses its effectiveness is a major reason for more accurate and efficient testing. 

The flow of the paper is difficult to follow. Putting the decision trees for deciding whether a specific case is probable or confirmed into the main text, rather than in supplementary materials, would be helpful.

---

## [Decision Letter · Decision Letter 2]

30 Jun 2022

Dear Dr. Ten Bosch,

Thank you for the submission of your revised Research Article "Evaluating and optimizing the use of diagnostics during epidemics: Application to the 2017 plague outbreak in Madagascar" for publication in PLOS Biology. On behalf of my colleagues and the Academic Editor, James Lloyd-Smith, I am pleased to say that we can in principle accept your manuscript for publication, provided you address any remaining formatting and reporting issues. These will be detailed in an email you should receive within 2-3 business days from our colleagues in the journal operations team; no action is required from you until then. Please note that we will not be able to formally accept your manuscript and schedule it for publication until you have completed any requested changes.

We suggest a change in the title that you could modify when the operations team contact you. Our suggestion is: "Analytical framework to evaluate and optimize the use of imperfect diagnostics during epidemics to inform public health responses".

PRESS

Sincerely, 

Paula

---

Senior Editor

PLOS Biology
